# Out-of-Distribution Validation for Bioactivity Prediction in Drug Discovery: Lessons from Materials Science

**Udit Surya Saha** [1] [*] , **Michele Vendruscolo** [1] , **Anne E. Carpenter** [2] , **Shantanu Singh** [2] , **Andreas Bender** [1] [3] ,
**Srijit Seal** [2] [*]

## Abstract

Recent advances in machine learning for materials science have significantly improved the prediction of novel materials. Building on these methods, we have adapted them for drug discovery, specifically focusing on assessing performance on out-of-distribution data. We found this approach more effective than conventional cross-validation methods by employing k-fold n-step forward cross-validation (SFCV) for predicting small molecules. Additionally, we introduced two new metrics: discovery yield and novelty error. These metrics provide deeper insights into model applicability and prediction accuracy for drug-like molecules. Based on our findings, we recommend incorporating these metrics into state-of-the-art bioactivity prediction models for drug discovery.

## 1. Introduction

Recently, many advancements have been made in developing computational methods for predicting properties in materials science. Suitable validation methods have also been introduced to estimate the performance of these predictive models [18, 3, 4]. Here, we investigated whether these validation methods can be translated into the field of drug discovery. Here, we address the problem of prospective validation. Since predictive models are trained and validated on the experimentally measured activity of libraries of compounds, real-world use in drug discovery requires strong performance on out-of-distribution data [11]. This is because the goal is often to accurately predict the properties of compounds that have not been synthesized yet. Inadequate prospective validation is a common issue in the drug discovery literature, often creating a mismatch between published

[*]Equal contribution [1]Yusuf Hamied Department of Chemistry, University of Cambridge, UK [2]Broad Institute of MIT and Harvard, Cambridge, MA, US [3]STAR-UBB Institute, Babeş-Bolyai University, Cluj-Napoca, Romania . Correspondence to: Srijit Seal <seal@broadinstitute.org>.

*Accepted at the 1st Machine Learning for Life and Material Sciences Workshop at ICML 2024.* Copyright 2024 by the author(s).

studies and real-world use [1, 2]. This problem is less severe in domains such as materials science, where the underlying physical principles are often known [4, 37], and protein folding, where evolution led to a lower-dimensional underlying space of possible solutions [6]. However, this problem is significant in drug discovery because the chemical space is vast (more than $10^{60}$ small molecules) and only explored to a limited extent, making it challenging to extrapolate to novel chemical series [1].

Benchmarking state-of-the-art models is more reliable for real-world decision-making when predicting compounds different from those in the training data space. However, most studies use cross-validation (CV) to evaluate models by randomly splitting the datasets for training versus testing [5]. This approach typically suffers from a limited applicability domain because test compounds are often similar to compounds in the training set. To mitigate this problem, splitting datasets by chemical scaffold or time-split has been proposed [14, 35, 8]. Even though these splits could be repeated for multiple external test sets (for example, repeated nested cross-validation), studies usually lack a detailed analysis of how variations in the drug discovery landscape and chemical space influence outcomes by differentiating between compounds unlikely to be drug-like and those that have desirable bioactivity and physicochemical properties.

To overcome these problems, one can take inspiration from machine learning studies for materials discovery, where validation and evaluation strategies have been developed for effective prospective discovery, i.e., identifying materials whose properties lie outside the range of training data [4, 34]. This trend makes sense because, in materials discovery, the goal is often to discover materials with a higher or lower property of interest (e.g., conductivity, band gap, etc.) than already known materials [3]. In one aspect, drug discovery is similar, as models are trained on data from previously known small molecules and then used to predict the bioactivity of compounds optimized to have desirable properties. Learning from these developments, we propose implementing a validation method and two metrics commonly used in prospective validation from materials science to the search for small molecules in drug discovery: (a) k-fold n-step forward cross-validation [34], (b) novelty error,

and (c) discovery yield [3].

During drug discovery, several properties of a compound are optimized simultaneously. One of the goals is to decrease logP, the logarithm of the partition coefficient (P) of a compound between n-octanol and water, a standard measure of hydrophobicity [10, 19]. Moderate logP values (typically between 1 and 3) are preferred in drug candidates to balance lipophilicity and hydrophilicity, enhancing oral bioavailability through good lipid membrane permeability and adequate aqueous solubility. A moderate logP value also ensures proper drug distribution, avoiding excessive accumulation in fatty tissues or insufficient penetration through cell membranes [17]. Therefore, we implemented a sorted k-fold n-step forward cross-validation (SFCV) to validate models, where the training and test datasets are selected based on continuous blocks of decreasing logP. When implementing SFCV, it is essential to ensure that the folds in the later iterations represent the desired logP values, which should be moderate (between 1 and 3). One could then assess whether a model fails to accurately predict compounds with desired bioactivity compared to other small molecules using discovery yield. Novelty error shows whether models can generalize on new, unseen data that differ significantly from the data on which the model was trained. This is similar to using the applicability domain [11] and distance to model measures [33]. Overall, we present these validation and evaluation metrics to the specific needs of toxicity and protein target prediction for small molecules [15].

## 2. Methods

### 2.1. Dataset

Models for predicting compound bioactivity require training datasets of activity readouts for many compounds. An activity readout is often expressed as an $IC_{50}$ value, the concentration at which a particular biological response is reduced to half (50%) of the original signal. While several datasets have binary readouts (active/inactive) for compounds towards given protein targets, these datasets are often noisy or employ arbitrary thresholds for binarising activity. Recently, it was demonstrated that combining data from different assay measurements is a significant noise source for such datasets [13]. Therefore, we restricted this study to having clean and single measurement type data, i.e., $IC_{50}$ values. Although the actual safety and potency of a compound depends on the dose and $C_{max}$ value (i.e., the maximum concentration in plasma in the organism) and is not inherent to the $IC_{50}$ of protein binding in a cell system, this study does not consider $C_{max}$ due to insufficient data in the public domain [25, 28]. Following previous studies, we selected the three relevant protein targets: hERG (1467 compounds), MAPK14 (1513 compounds), and VEGFR2 (1751 compounds) from Landrum et al. [13]. hERG in-

hibition is well-known to cause cardiotoxicity [26, 7] and should be avoided in drug discovery. The other two targets represent positive attributes: MAPK14 inhibitors could potentially treat multiple diseases, such as neurodegenerative diseases, cardiovascular cases, and cancer. VEGFR2 is a potent therapeutic target for treating angiogenesis-related tumors. From the literature, we found that approved drugs have an $IC_{50} > 6.3$ $\mu$M for hERG (less likely to inhibit hERG and cause toxicity) [32]. On the other hand, drugs that target the other two proteins would ideally have $IC_{50} < 100$ nM for VEGFR2 [36] or MAPK14 inhibition [21].

We converted all $IC_{50}$ values into $pIC_{50}$ values, the negative logarithm in base 10 of the $IC_{50}$ value expressed in molar concentration (M). The $pIC_{50}$ value provides a more intuitive understanding of compound potency: higher $pIC_{50}$ values indicate greater potency, as they correspond to lower $IC_{50}$ concentrations [30].

### 2.2. Compound Featurization and Model Algorithm

Compound SMILES were first standardized (for details, see Appendix 5.1). To represent compounds as features, we used 2048-bit ECFP4 fingerprints (also known as Morgan fingerprints, as implemented in RDKit), which encode chemical structural features in a binary vector format, and logP values were calculated using RDKit. We used three models: (a) Random Forest (RF) Regressor, (b) Gradient Boosting, and (c) Multi-Layer Perceptrons (MLP, as implemented in scikit-learn version 1.3.0 [23] for all the analysis in this paper) to use molecular fingerprints encoded in a feature matrix to predict chemical properties of interest, namely the pIC50 values. The Random Forest Regressor was coded dynamically to set the number of trees based on the training data size, using the square root of the number of samples but not exceeding 25 trees. A similar approach was implemented for the other models, with the number of estimators in Gradient Boosting and the number of hidden-layer nodes in the MLP being limited to 25. This approach balances model complexity, helping to prevent overfitting for these tasks where data is limited.

### 2.3. Model Training and Evaluation

To validate models, we adopted a step-forward cross-validation (SFCV) method [34] (as shown in Figure 1) and compared it with conventional k-fold cross-validation (CV). To implement the SFCV method, we sorted the dataset from high to low logP values and divided the dataset into ten bins. For the first iteration, we used the first bin for training and the second for testing. In each successive iteration, we expanded the training set by adding the next bin while using the subsequent bin (with lower logP compounds) for testing. This method mimics a real-world scenario where chemical structures undergo optimization to become more

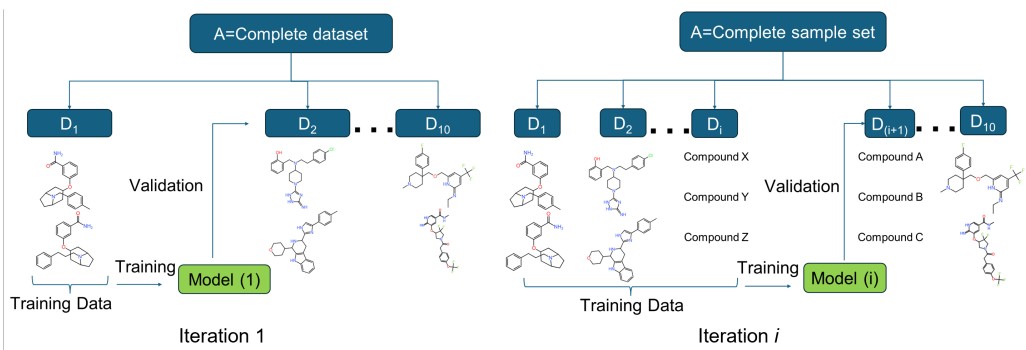

*Figure 1.* Workflow for 10-fold step-forward cross-validation (SFCV). The $D_i$ dataset block can be sorted via a calculated or experimental molecular property. Here, we used logP.

drug-like [29]. To explore the impact of logP sorting on model performance, we also implemented an SFCV without sorting, effectively creating a random step forward cross-validation. This setup allowed us to assess how sorting training data influences predictive accuracy and the discovery of more drug-like molecules. SFCV, however, leads to lesser training data for each iteration. To explore the impact of dataset size, we implemented a conventional 10-fold CV with random splits for a direct comparison, with one of the ten randomly selected folds as a test set for each turn, and the remaining folds composed of the training set. To explore the impact of chemical scaffolds, we further implemented a 10-fold CV with scaffold-based splitting (as implemented in ScaffoldSplitter in DeepChem version 2.7.1 [20, 31]), where molecules are grouped based on the Bemis-Murcko scaffold representation. The predictive performance for each iteration was evaluated by calculating the Root Mean Square Error (RMSE, a point-based measure of model accuracy) and $R^2$ (an overall-fit measure of model accuracy).

### 2.4. Discovery Yield and Novelty Error

We introduced two metrics to assess the model in a real-world context and adopted them from machine learning in material science: discovery yield and novelty error rates [3]. To define discovery yield, first, we identified highly potent and safe compounds called discovery compounds. For hERG, compounds with a $pIC_{50}$ value lower than 5.2 are defined as discovery compounds [32]. Similarly, for VEGFR2 and MAPK14, compounds with a pIC50 value exceeding 7.0 are defined as discovery compounds [36, 21]. We defined discovery yield as the fraction of discovery compounds for which the $IC_{50}$ value was predicted within an error range of 0.5 log units. Furthermore, for each test set, we analyzed the performance specifically for compounds structurally dissimilar to those in the training dataset (<0.55 Tanimoto similarity for the nearest neighbor [9]). We defined the mean absolute error on these compounds as novelty error.

## 3. Results and Discussion

In the work, we evaluated the effectiveness of sorted step-forward cross-validation (SFCV) in predicting compounds' bioactivity and toxicity. We sorted compounds based on logP values and used an SFCV to assess whether the models could accurately predict desired bioactivity and generalize to new, unseen data. We selected three protein targets relevant to drug discovery: hERG, MAPK14, and VEGFR2 datasets, comprising 1,262, 1,445, and 1,641 unique compounds. Available data for the $pIC_{50}$ of compounds against these protein targets (a measure of potency) show a wide distribution of biological activity values (Figure S1). The distribution of $pIC_{50}$ values for training and test datasets remained relatively similar for sorted and unsorted SFCV, indicating that sorting by logP does not significantly affect the activity distribution within the datasets (Figure S1).

### 3.1. Sorted SFCV selected more novel compounds than CV methods

Next, we analyzed the chemical space selected for training/testing splits using four validation strategies (Figure S2). Sorted SFCV selected test compounds with progressively lower logP values, as expected. In contrast, other validation methods did not show this trend—cross-validation (CV) with scaffold-based splits selected compounds in different molecular weight spaces in earlier batches. However, in later batches, the distribution of test compounds resembled that of random splitting. This can be attributed to the nature of scaffold splits: larger groups of molecules remaining in later folds are often very dissimilar to each other and thus not representative of any specific scaffold type (that is, chemistry datasets usually do not form exact clusters of molecules, making it challenging to split them neatly into scaffold clusters). As a result, scaffold-based CV effectively identifies some groups of scaffolds in the early batches but also groups some relatively dissimilar compounds in later batches.

To determine which validation is likely to test compounds dissimilar to the training data, we analyzed each method using the hERG prediction task as an example (Figure 2a). Sorted and unsorted SFCVs showed more dissimilar compounds in early testing batches, which gradually decreased. CV with random splitting maintained a much lower number of dissimilar test compounds across all batches. CV selected fewer dissimilar compounds to test than sorted SFCV (170 with sorted SFCV compared to 127 compounds with scaffold splits and 24 with random splits; Figure S3a). Thus, sorted SFCV is better for evaluating compounds with novel structures.

### 3.2. Sorted SFCV selected more discovery compounds than CV methods

We compared the validation methods based on how they split between train and test data in the biological space and how models could be evaluated. All validation methods selected discovery compounds (compounds with $pIC_{50} < 5.2$ in the case of hERG) in the test folds (Figure S3b). Using unsorted SFCV and CV (random splits), models predicted a higher number within an error range of 0.5 log units in early batches but fluctuated in later batches (Figure 2b). Models evaluated using sorted SFCV showed an increasing trend in accurately predicted discovery compounds towards later batches, where compounds have a desirable lower logP.

Among these discovery compounds, there were also novel compounds structurally dissimilar to those in the training data ($T_c < 0.55$). Unsorted and sorted SFCVs effectively selected these dissimilar discovery compounds as test compounds (Figure 2c). CV with scaffold split had fewer such compounds than SFCV methods (29 in CV with scaffold splits vs. 59 in sorted SFCV and 66 in unsorted SFCV in the hERG task). CV with random splits identified very few dissimilar discovery compounds (only 2 in the hERG task), shown in Figure S3c. We checked how many predictions for these dissimilar discovery compounds were within an error range of 0.5 log units. The pattern was similar to the number of compounds detected: most methods could identify these compounds (Figure 2d). However, as SFCV had more such compounds to predict, it was advantageous to evaluate models using this validation strategy.

Overall, sorted SFCV balances identifying diversely structured compounds with consistent and improved predictions over testing batches. CV (random split) maintains an optimistic approach with stable performance, while unsorted SFCV and CV (scaffold split) offer higher initial diversity but may lack consistency in later batches.

### 3.3. Sorted SFCV selects more discovery compounds

We compared the validation methods based on how they split train/test data in the biological space and how well

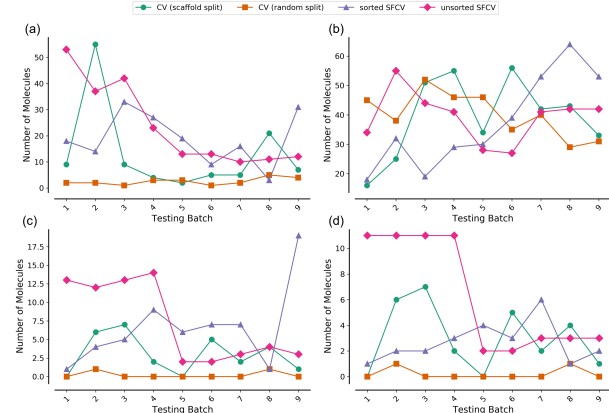

*Figure 2.* (a) Number of compounds dissimilar to training data ($T_c < 0.55$), (b-d) Number of discovery compounds ($pIC_{50} < 5.2$) in the test set predicted within an error range of 0.5 log unit (b), and discovery compounds dissimilar to training data ($T_c < 0.55$) (c), and discovery compounds dissimilar to training data ($T_c < 0.55$) correcting predicted within an error range of 0.5 log unit (d), as shown for the hERG target prediction task across four validation methods of sorted SFCV, unsorted SFCV, cross-validation with random splits, and cross-validation with scaffold splits (for each of the first nine test folds). $T_c$: Tanimoto Similarity; SFCV: Step-Forward Cross-Validation.

models could be evaluated in that space. Unsorted SFCV and CV (random split) achieved a higher number of discovery compounds (compounds with $pIC_{50} < 5.2$ in the case of hERG) predicted within an error range of 0.5 log units in early batches, but this fluctuated in later batches (Figure 2b). Sorted SFCV showed an increasing trend in accurately predicted discovery compounds towards later batches.

Among these total discovery compounds, there were also novel compounds structurally dissimilar to those in the training data (Tc < 0.55). Unsorted and sorted SFCVs effectively selected these dissimilar discovery compounds as test compounds (Figure 2c). CV (scaffold split) had fewer such compounds compared to SFCV methods (29 vs. 59 in sorted SFCV, 66 in unsorted SFCV), while CV (random split) identified very few (only 2 in the hERG task). We checked predictions within an error range of 0.5 log units for these dissimilar discovery compounds. The pattern was similar to the total number of compounds detected: most methods could identify these compounds (Figure 2d). However, as sorted SFCV had more such compounds to predict, it is easier to evaluate models using those validation strategies.

Overall, sorted SFCV balances identify diversely structured compounds with consistent and improved predictions over testing batches. CV (random split) maintains an optimistic approach with stable performance, while unsorted SFCV and CV (scaffold split) offer higher initial diversity but may lack consistency in later batches.

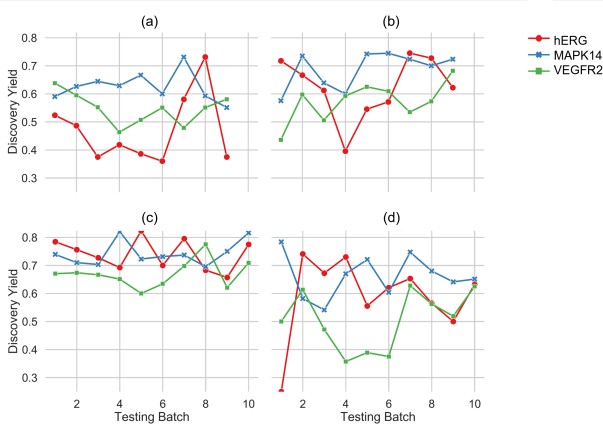

*Figure 3.* Discovery yield for models validated using (a) sorted SFCV, (b) unsorted SFCV, (c) cross-validation with random splits, and (d) cross-validation with scaffold splits for all three protein targets when using a Random Forest model. SFCV: Step-Forward Cross-Validation.

### 3.4. Extrapolation error: Sorted SFCV shows that predicting bioactivity for compounds with low lipophilicity is challenging

Conventional CV with random splitting exhibited better performance metrics (mean RMSE=0.58±0.04 for RF) compared to sorted SFCV (mean RMSE=0.77±0.03 for RF) across all protein targets (Table 1). However, all the models evaluated, including RF, xgboost, and MLP, demonstrated limited ability to extrapolate when predicting for compounds with lower or higher $pIC_{50}$ values than remaining compounds (Figure S4 for VEGFR2). The extrapolation challenges are less noticeable under unsorted SFCV or conventional CV conditions than under sorted SFCV. Sorted SFCV (Figure S4a) results in significantly higher absolute errors for compounds with the highest $pIC_{50}$ values—up to 100 times the experimental IC50 value across multiple test compounds—compared to errors obtained using alternative validation methods. These results indicate that evaluating models is more difficult for compounds with lower logP than the training data. It will thus be essential to develop models capable of adapting to desirable regions of chemical space. For example, in the first 5 out of 10 iterations of the hERG dataset, the models performed poorly when validated using sorted SFCV (mean $R^2$=0.22) compared to CV with random splits (mean $R^2$=0.74), where more data were available to train individual models and the chemical space of test data was more similar to training datasets (Figure S5, see Table S1 for more details). Overall, sorted SFCVs can better evaluate models, considering the inherent challenges of model extrapolation to less-represented regions of the chemical space.

### 3.5. Discovery Yield: Sorted SFCV performs similarly to CV with scaffold split but has a more systematic approach to selecting compounds.

We next evaluated the models based on their discovery yield. Unlike accuracy, which measures the overall correctness of predictions across all compounds, discovery yield focuses on the ability of the model to predict activity for compounds with desired properties—such as low toxicity for hERG and high potency for MAPK14 and VEGFR2, as compounds selected in this way offer the most potential for further development. The discovery yield remained high (0.60 to 0.82) across all iterations of conventional CV with random splits (Figure 3). However, this metric showed notable fluctuations under unsorted SFCV (0.40 to 0.75), CV with scaffold split (0.25 to 0.78), and sorted SFCV (0.36 to 0.73). These fluctuations within each method can be attributed to the inherent differences in the composition of the test datasets. In sorted SFCV, the test dataset tends to be dissimilar to the training data. This difference creates a more challenging test environment that mirrors real-world scenarios in drug discovery.

Conversely, in conventional CV with random splits, the test sets mirror the training data distribution more closely, making predictions for compounds relatively easier. In many drug projects, properties such as logP often increase as optimization progresses in Design-Make-Test-Analyze (DMTA) cycles, which is undesirable[16, 24]. A conventional CV with random splits may appear to provide stable performance metrics, but it gives a misleading representation of a model's efficacy in practical settings. On the other hand, CV with scaffold splits represents a stricter task. Although this simulates the challenges encountered in real-world applications, a systematic approach is needed to introduce novel scaffolds. It can sometimes be too random, introducing wide variability and unpredictability in the performance metrics over various folds. Overall, our results indicate that sorted SFCV is a better validation approach that simulates the progression in molecule optimization.

### 3.6. Novelty Error is more consistent for sorted SFCV

A key question about models aimed at predicting the activity of small molecules is their applicability domain - that is, in what regions of chemical space does a model offer accurate predictions?[11, 27] We explored this question by assessing the novelty error, defined as the mean absolute error for compounds in the test dataset that are structurally distinct from the training set across each iteration. We observed that the novelty error remains low throughout the iterations of sorted SFCV (0.88 ± 0.21). Meanwhile, it fluctuates more for unsorted SFCV (0.81 ± 0.31) and conventional cross-validation with random splits (0.80 ± 0.41) or with scaffold-splits (0.80 ± 0.48) (Figure 4). One might expect

*Table 1.* Mean performance metrics in predicting pIC50 over the three protein targets when using RF models for each validation technique used in this study. RF: Random Forest; SFCV: Step-Forward Cross-Validation.

| Validation | $R^2$ (mean ± std) | RMSE (mean ± std) | Novelty Error (mean ± std) | Discovery Yield (mean ± std) |
|---|---|---|---|---|
| CV (Random) | 0.71 ± 0.05 | 0.58 ± 0.04 | 0.80 ± 0.09 | 0.72 ± 0.04 |
| CV (Scaffold) | 0.49 ± 0.11 | 0.71 ± 0.05 | 0.80 ± 0.09 | 0.59 ± 0.08 |
| SFCV (Sorted) | 0.43 ± 0.11 | 0.77 ± 0.03 | 0.88 ± 0.04 | 0.55 ± 0.08 |
| SFCV (Unsorted) | 0.59 ± 0.08 | 0.68 ± 0.02 | 0.82 ± 0.15 | 0.63 ± 0.06 |

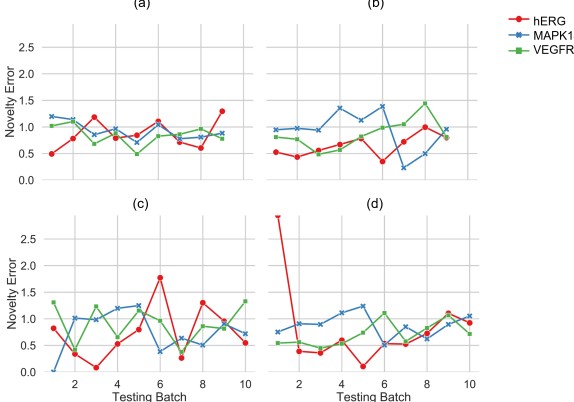

*Figure 4.* Novelty error for models validated using (a) sorted SFCV, (b) unsorted SFCV, (c) cross-validation with random splits, and (d) cross-validation with scaffold splits for all three protein targets when using a Random Forest model. SFCV: Step-Forward Cross-Validation.

that sorted SFCV would increase the difficulty of accurate prediction, thereby raising the error rates. However, the consistency in lower mean errors during sorted SFCV can be explained by the nature of novelty in this validation set. In sorted SFCV, the novelty of compounds is defined more narrowly, focusing on compounds that have not been seen but are systematically selected based on a specific biological property (here, logP). This systematic approach ensures that while the compounds are novel, they are not randomly diverse[22], which leads to more predictable error patterns.

In contrast, in unsorted SFCVs and CVs, the novelty is broader; random splits introduce a wider variety of compounds into the training set, and novel compounds in the test set need to be further out-of-distribution and highly varied, making predictions for these compounds more challenging and error-prone. We demonstrate that sorted SFCV can minimize overfitting in random chemical spaces and reduce novelty error.

## 4. Conclusion

We have investigated the potential of applying machine learning validation techniques developed in materials sci-

ence to molecular property prediction in drug discovery. We made a conceptual argument that k-fold n-step forward cross-validation (SFCV) [34] better matches real-world use and is more stringent than conventional k-fold cross-validation (CV). We show that sorted SFCV is better at selecting novel structures where the biological activity is of interest compared to the second-best validation approach of CV with scaffold-based splitting. We have also translated discovery yield [3] and novelty error [3] into the drug discovery field as metrics to evaluate models to accurately predict properties of compounds that diverge from the training data in terms of biological property and chemical structure.

Using a sorted SFCV, we have demonstrated that it is challenging for any model to perform consistently in a varying chemical space that systematically changes towards a desired property, such as logP. The differing discovery yields observed across different validation methods highlight the influence of dataset composition and model training strategies on predictive accuracy. Models trained using a sorted SFCV are less prone to overfitting and yield lower novelty errors. In the future, models optimized using sorted SFCV—implemented in DMTA cycles [24]—could potentially generalize better for diverse compounds. In the future, SFCV could be implemented for other activity measures, such as inhibition constants ($K_i$) and potency measures from ChEMBL, offering additional insights. Various activity measures are crucial for understanding compound-protein interactions in drug discovery, and findings from other activity measures could help elucidate how they relate to $IC_{50}$, the property explored in the current work.

Overall, we recommend the techniques presented in this study to align model testing with the directional nature of drug discovery, where compounds are gradually optimized to become more drug-like [1]. Our results suggest that integrating machine learning strategies developed in materials science into drug discovery pipelines offers additional opportunities for a more accurate assessment of model performance.

## Code Release

An example notebook implementation for sorted SFCV is available at https://github.com/srijitseal/ValidationDiscovery-/MolProp/tree/main/local_implementation_notebook. All code for this work is publicly released on https://github.com/srijitseal/ValidationDiscovery-MolProp.

## Data Release

All datasets used in this paper have been publicly made available on https://github.com/srijitseal/ValidationDiscovery-MolProp/tree/main/data

## Funding

The authors acknowledge funding from the National Institutes of Health R35 GM122547 (to AEC).

## Acknowledgments

The authors thank Jessica D. Ewald (Broad Institute of MIT and Harvard) for her comments on an earlier version of this manuscript.

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

# 5. Appendix

## 5.1. Compound Standardization

We used the RDKit library (version 2023.9.427, for all the analysis in this paper) to standardize the SMILES (Simplified Molecular Input Line Entry System) representations of chemical structures. Multiple standardization steps included RDKit cleanup functions to desalt and reionize the molecules and neutralize charges. Further standardization involved normalizing isotopes, stereochemistry, and tautomers through canonicalization methods implemented in the RDKit MolStandardize module [12]. Each molecule was iteratively processed up to five times until the output SMILES stabilized. Otherwise, the most frequently occurring standardized SMILES was selected as the final output. For each unique SMILES, the median pIC50 values of replicates were calculated to summarize the central tendency and minimize the impact of outliers in the data. Finally, this resulted in 1262 unique compounds for the hERG dataset, 1445 for the MAPK14 dataset, and 1641 for the VEGFR2 dataset.

## 5.2. Tables

*Table S1.* Performance metrics for predicting the pIC50 for three protein targets in this work for each combination of models and validation techniques used in this study. Cross-validation (CV) is shown to perform better than other techniques.

| Target | Method | Validation | R2 (mean ± std) | RMSE (mean ± std) | Novelty Error (mean ± std) | Discovery Yield (mean ± std) |
|---|---|---|---|---|---|---|
| hERG | MLP | CV (Random) | 0.58 ± 0.10 | 0.66 ± 0.06 | 1.28 ± 0.34 | 0.70 ± 0.11 |
| hERG | MLP | CV (Scaffold) | 0.08 ± 0.58 | 0.81 ± 0.21 | 1.07 ± 0.59 | 0.57 ± 0.07 |
| hERG | MLP | SFCV (Sorted) | 0.07 ± 0.65 | 0.85 ± 0.25 | 0.99 ± 0.37 | 0.61 ± 0.11 |
| hERG | MLP | SFCV (Unsorted) | 0.26 ± 0.47 | 0.79 ± 0.16 | 0.80 ± 0.31 | 0.64 ± 0.10 |
| hERG | RF | CV (Random) | 0.70 ± 0.07 | 0.55 ± 0.07 | 0.74 ± 0.51 | 0.74 ± 0.06 |
| hERG | RF | CV (Scaffold) | 0.41 ± 0.31 | 0.69 ± 0.35 | 0.82 ± 0.80 | 0.59 ± 0.14 |
| hERG | RF | SFCV (Sorted) | 0.32 ± 0.46 | 0.73 ± 0.20 | 0.87 ± 0.27 | 0.47 ± 0.12 |
| hERG | RF | SFCV (Unsorted) | 0.52 ± 0.18 | 0.66 ± 0.09 | 0.65 ± 0.20 | 0.62 ± 0.11 |
| hERG | xgboost | CV (Random) | 0.70 ± 0.09 | 0.55 ± 0.07 | 0.72 ± 0.49 | 0.71 ± 0.10 |
| hERG | xgboost | CV (Scaffold) | 0.38 ± 0.27 | 0.71 ± 0.33 | 0.80 ± 0.82 | 0.58 ± 0.10 |
| hERG | xgboost | SFCV (Sorted) | 0.35 ± 0.43 | 0.71 ± 0.18 | 0.84 ± 0.32 | 0.45 ± 0.14 |
| hERG | xgboost | SFCV (Unsorted) | 0.51 ± 0.22 | 0.66 ± 0.09 | 0.60 ± 0.25 | 0.63 ± 0.11 |
| MAPK14 | MLP | CV (Random) | 0.66 ± 0.07 | 0.67 ± 0.05 | 0.67 ± 0.66 | 0.65 ± 0.05 |
| MAPK14 | MLP | CV (Scaffold) | 0.39 ± 0.41 | 0.83 ± 0.12 | 1.04 ± 0.45 | 0.54 ± 0.07 |
| MAPK14 | MLP | SFCV (Sorted) | 0.39 ± 0.21 | 0.88 ± 0.17 | 1.05 ± 0.40 | 0.47 ± 0.07 |
| MAPK14 | MLP | SFCV (Unsorted) | 0.56 ± 0.18 | 0.76 ± 0.14 | 0.91 ± 0.55 | 0.59 ± 0.11 |
| MAPK14 | RF | CV (Random) | 0.76 ± 0.04 | 0.57 ± 0.05 | 0.76 ± 0.39 | 0.74 ± 0.04 |
| MAPK14 | RF | CV (Scaffold) | 0.62 ± 0.18 | 0.67 ± 0.08 | 0.88 ± 0.22 | 0.66 ± 0.08 |
| MAPK14 | RF | SFCV (Sorted) | 0.54 ± 0.09 | 0.77 ± 0.11 | 0.93 ± 0.17 | 0.63 ± 0.05 |
| MAPK14 | RF | SFCV (Unsorted) | 0.67 ± 0.09 | 0.67 ± 0.10 | 0.93 ± 0.37 | 0.69 ± 0.06 |
| MAPK14 | xgboost | CV (Random) | 0.74 ± 0.05 | 0.59 ± 0.06 | 0.79 ± 0.50 | 0.73 ± 0.04 |
| MAPK14 | xgboost | CV (Scaffold) | 0.53 ± 0.26 | 0.73 ± 0.10 | 0.95 ± 0.13 | 0.60 ± 0.08 |
| MAPK14 | xgboost | SFCV (Sorted) | 0.48 ± 0.12 | 0.81 ± 0.12 | 0.98 ± 0.19 | 0.61 ± 0.06 |
| MAPK14 | xgboost | SFCV (Unsorted) | 0.64 ± 0.09 | 0.70 ± 0.09 | 1.03 ± 0.34 | 0.67 ± 0.08 |
| VEGFR2 | MLP | CV (Random) | 0.53 ± 0.06 | 0.74 ± 0.06 | 1.04 ± 0.59 | 0.59 ± 0.03 |
| VEGFR2 | MLP | CV (Scaffold) | 0.27 ± 0.21 | 0.89 ± 0.11 | 0.87 ± 0.34 | 0.44 ± 0.08 |
| VEGFR2 | MLP | SFCV (Sorted) | 0.04 ± 0.35 | 1.02 ± 0.13 | 1.33 ± 0.17 | 0.39 ± 0.06 |
| VEGFR2 | MLP | SFCV (Unsorted) | 0.43 ± 0.16 | 0.81 ± 0.10 | 0.99 ± 0.37 | 0.48 ± 0.09 |
| VEGFR2 | RF | CV (Random) | 0.66 ± 0.04 | 0.63 ± 0.05 | 0.91 ± 0.35 | 0.67 ± 0.05 |
| VEGFR2 | RF | CV (Scaffold) | 0.46 ± 0.19 | 0.77 ± 0.08 | 0.71 ± 0.23 | 0.50 ± 0.10 |
| VEGFR2 | RF | SFCV (Sorted) | 0.43 ± 0.13 | 0.80 ± 0.07 | 0.84 ± 0.18 | 0.55 ± 0.06 |
| VEGFR2 | RF | SFCV (Unsorted) | 0.58 ± 0.08 | 0.70 ± 0.06 | 0.86 ± 0.28 | 0.57 ± 0.07 |
| VEGFR2 | xgboost | CV (Random) | 0.62 ± 0.03 | 0.67 ± 0.04 | 0.95 ± 0.28 | 0.59 ± 0.07 |
| VEGFR2 | xgboost | CV (Scaffold) | 0.41 ± 0.26 | 0.79 ± 0.10 | 0.71 ± 0.21 | 0.47 ± 0.11 |
| VEGFR2 | xgboost | SFCV (Sorted) | 0.41 ± 0.12 | 0.82 ± 0.07 | 0.88 ± 0.17 | 0.55 ± 0.06 |
| VEGFR2 | xgboost | SFCV (Unsorted) | 0.54 ± 0.07 | 0.74 ± 0.04 | 0.86 ± 0.19 | 0.55 ± 0.05 |

## 5.3. Figures

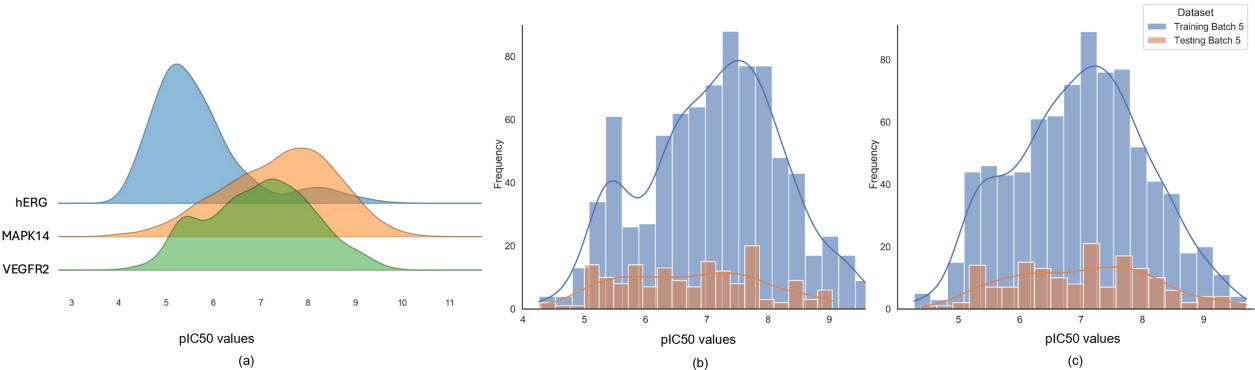

*Figure S1.* (a) Distribution of biological activity (pIC50 values) for the three protein target datasets. (b,c) Distribution of $pIC_{50}$ values in the training set and test set of the 5th iteration for sorted SFCV (b) and unsorted SFCV (c). SFCV: Step-Forward Cross-Validation

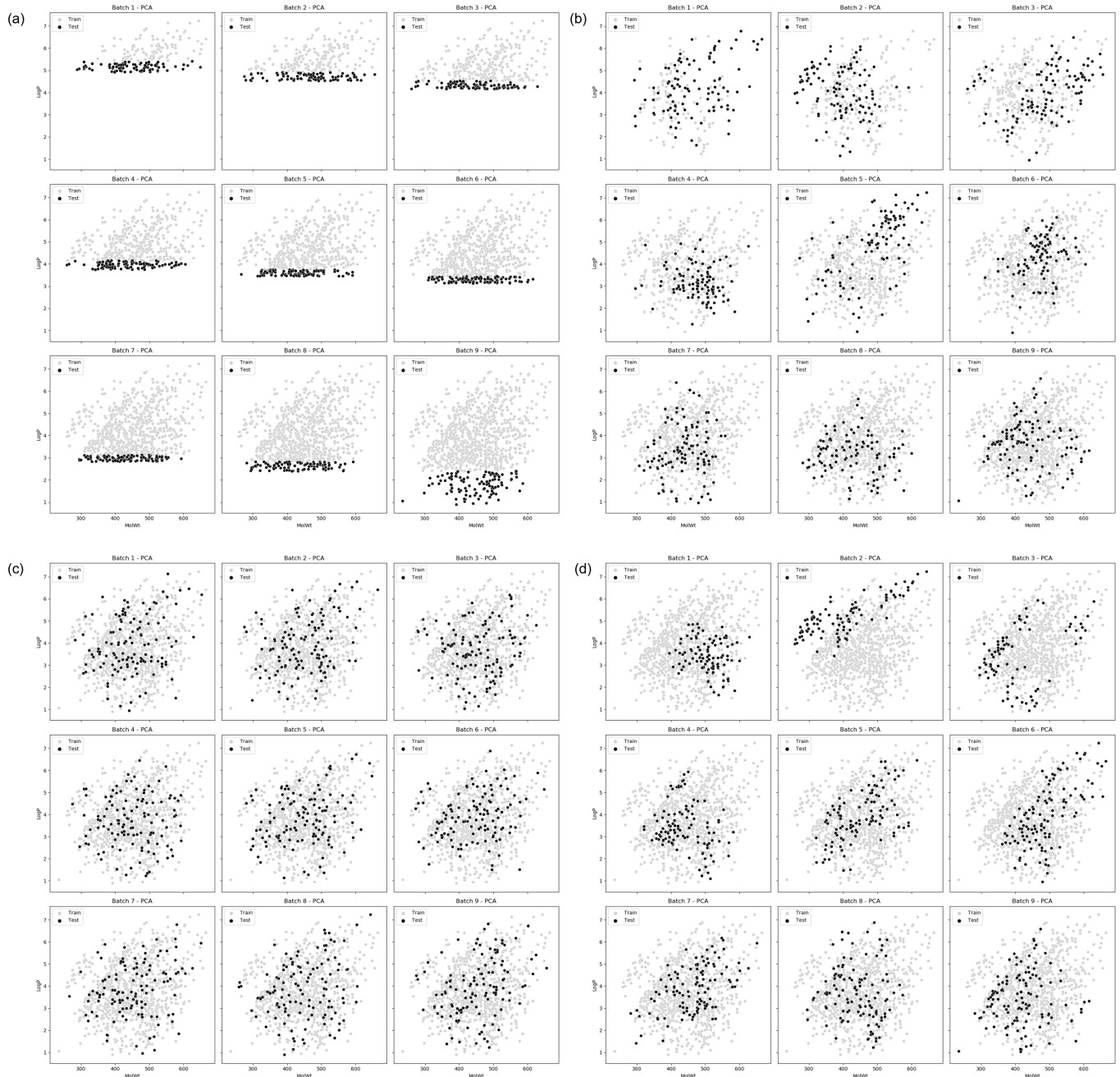

*Figure S2.* Comparison of logP and Molecular Weight for physico-chemical space for compounds selected as training and test sets across various iterations for the hERG target prediction task for (a) sorted SFCV, (b) unsorted SFCV, (c) cross-validation with random splits, and (d) cross-validation with scaffold splits for the (first) nine iterations for Random Forest models. SFCV: Step-Forward Cross-Validation

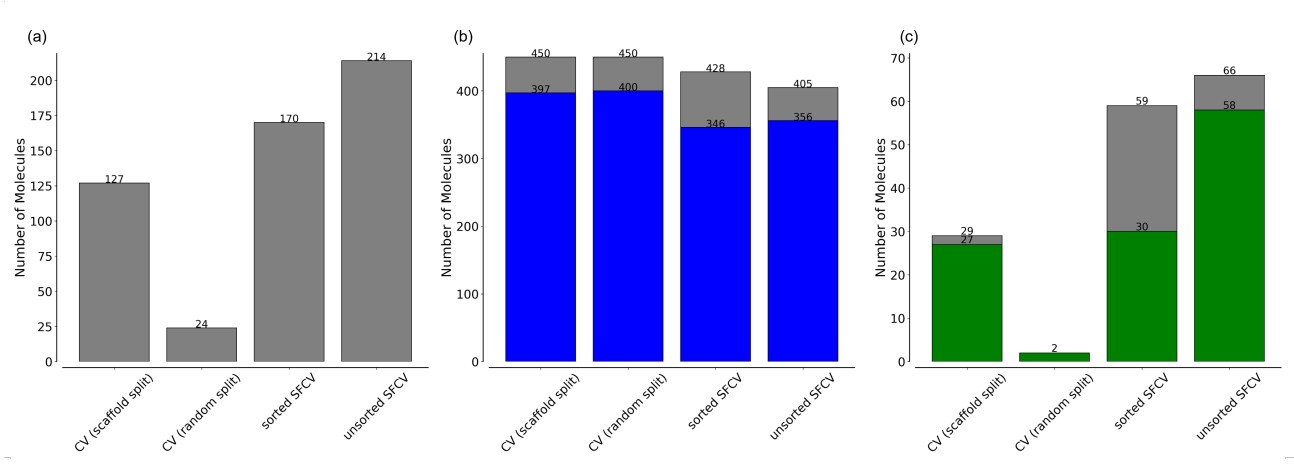

*Figure S3.* (a) The total number of compounds dissimilar to training data ($T_c < 0.55$), (b) the total number of discovery compounds ($pIC_{50} < 5.2$) in the test set predicted within an error range of 0.5 log unit, and (c) the total discovery compounds dissimilar to training data ($T_c < 0.55$), as shown for the hERG target prediction task across four validation methods of sorted SFCV, unsorted SFCV, cross-validation with random splits, and cross-validation with scaffold splits (combined for all test folds). The colored stack shows how many predictions are within a range of 0.5 log fold unit error. $T_c$: Tanimoto Similarity; SFCV: Step-Forward Cross-Validation.

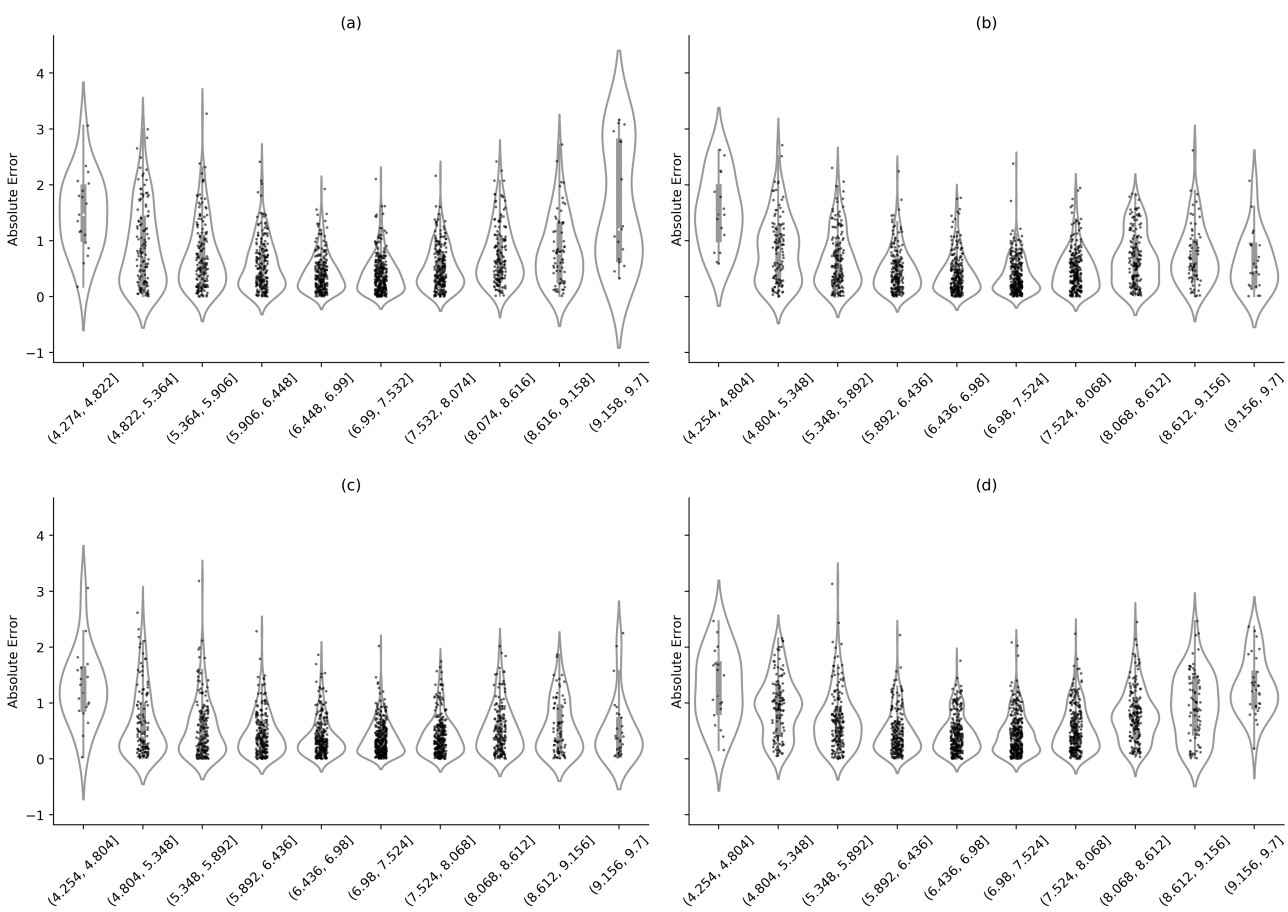

*Figure S4.* Absolute Error for VEGFR2 target prediction (sorted by pIC50 values) for (a) sorted SFCV, (b) unsorted SFCV, (c) cross-validation with random splits, and (d) cross-validation with scaffold splits. SFCV: Step-Forward Cross-Validation

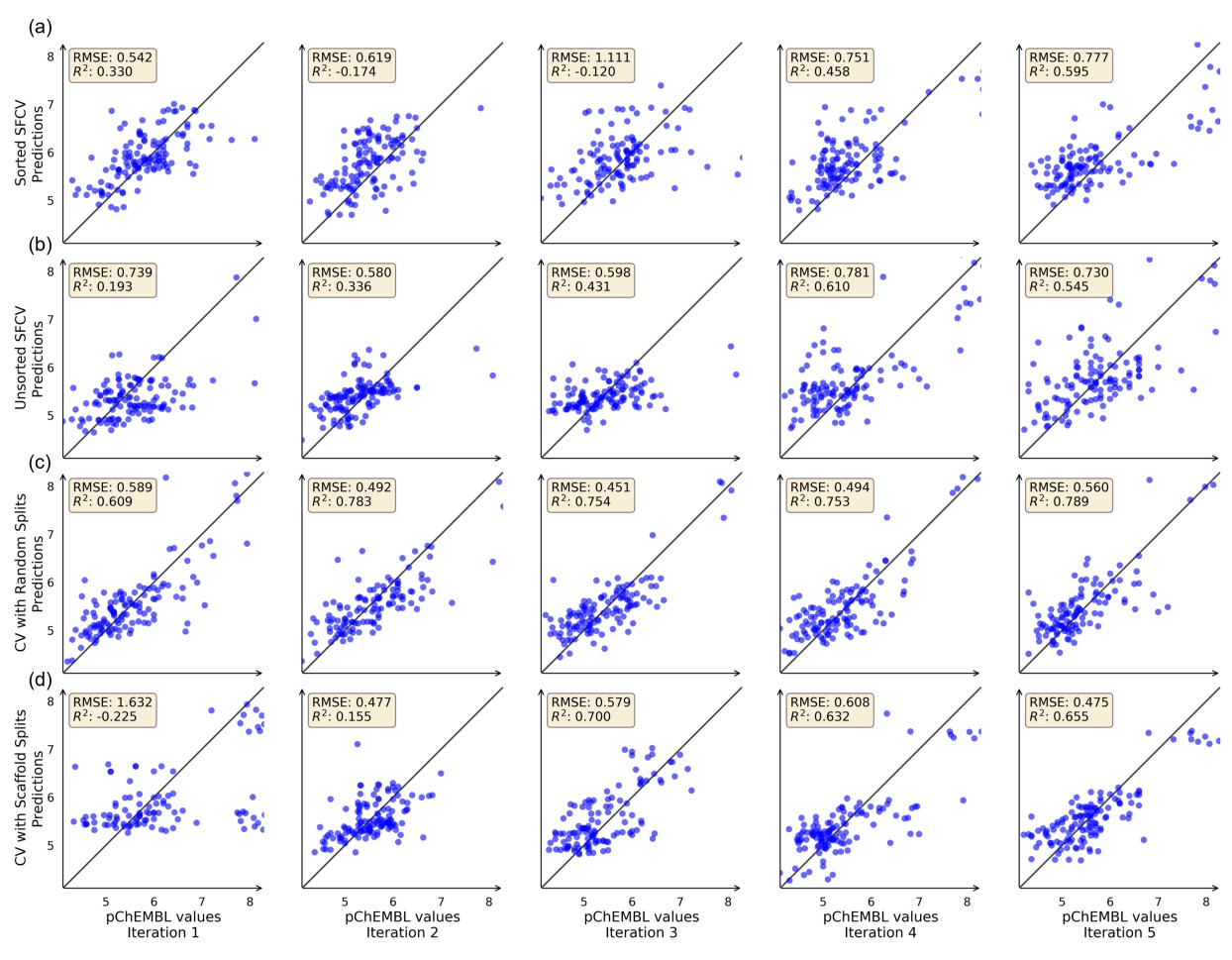

*Figure S5.* Parity plots for hERG target prediction for (a) sorted SFCV, (b) unsorted SFCV, (c) cross-validation with random splits, and (d) cross-validation with scaffold splits for the first five iterations for Random Forest models. SFCV: Step-Forward Cross-Validation