# OpenReview forum: "Out-of-Distribution Validation for Bioactivity Prediction  in Drug Discovery: Lessons from Materials Science"
_ICML.cc/2024/Workshop/ML4LMS — ML4LMS Poster_

### Official Review · Reviewer_HCwy · 2024-06-02
**Great findings and contributions that would benefit from a few revisions**

**Rating:** 6
**Confidence:** 4

**Review:**

I thoroughly enjoyed reading your paper and found it to be very well-written with a clear and well-structured methodology. The research presents significant insights into the field, and the results are presented in an organized and comprehensible manner. However, I believe there are certain aspects that could be improved or further explored to enhance the impact of your findings. Below are my detailed comments and suggestions.

Strengths:
1. Clear Methodology: The paper provides a clear and concise explanation of the methodology and background, which makes it easy to follow and understand. The steps are well-documented and the rationale behind each choice is well-articulated.
2. Presentation: Paper is well-organized and results are presented in a logical sequence.  The use of figures and tables is appropriate and aids in the understanding of the points the authors are making.

Suggestions for Improvement:
1. Parameter Datasets: I believe this paper would greatly benefit from exploring different parameter datasets. For instance, considering Ki (inhibition constants) could provide additional insights as they are quite similar to IC50 and are crucial in compound-protein interactions in the context of drug discovery. Expanding the scope of parameters might reveal new dimensions of your findings and make the study more comprehensive.
2. Novelty Error Metric: While the concept of 'novelty error' is interesting, I don't believe it is a novel metric or method introduced solely in this paper. Several previous studies have utilized similar validation metrics, though you may be the first to formally name it. A brief literature review acknowledging these previous works could provide better context and strengthen the justification for its use.
3. Additional Metrics: Including additional statistical measures with your results such as Pearson's R and Spearman's rho would enhance the understanding of the ranking effects in your study.

Questions and Clarification:
1. How does SFCV lead to moderate logP values? The results section does not clearly support this claim. Additional explanation of supporting data would be beneficial to substantiate this point.
2. For the results presented in Figure S2, have you considered using similarity scores such as Tanimoto coefficients instead of PCA-based analyses? This could provide a different perspective on the similarity relationships within your dataset.
3. In Figure S2, are you sorting the scaffold-based CV as well? Its unclear why the novelty of the compounds should change with increasing batches if the compounds are not sorted. Clarification on this point would help readers understand this claim and the figures better.
4. Have you evaluated your metrics using an alternative approach to training, such as running training and validation on multiple data seeds instead of CV? This could provide a robustness check for your findings and potentially highlight the stability of your work.

---

### Official Review · Reviewer_SxKq · 2024-06-12
**This is an application of a method to estimate success in extrapolation when exploring materials/chemical compounds. Greater dissemination of these ideas could be useful in this area.**

**Rating:** 7
**Confidence:** 2

**Review:**

This work brings to drug target prediction a method of generalized cross-validation proposed in the context of materials science by Xiong et al. 2020. If the purpose of our ML efforts is to find 'better' materials, for example, substances with higher superconducting transition temperatures $T_c$, we might get a better estimate of our performance by training on a part of data with lower $T_c$ and testing on those with higher $T_c$. Step-forward cross-validation (SFCV) of Xiong et al. seems to be based on this idea. The authors organize their training data by decreasing logP and use SFCV, employing some novel metrics for performance. they claim this procedure benefits drug discovery.

One thing that is not clear to me is that, since the desired logP values seem to be some moderate values (between 1 and 3 according to the manuscript),  whether going from higher to lower logP is a good idea or not.